# Intestinal Permeation Enhancers for Oral Delivery of Macromolecules: A Comparison between Salcaprozate Sodium (SNAC) and Sodium Caprate (C_10_)

**DOI:** 10.3390/pharmaceutics11020078

**Published:** 2019-02-13

**Authors:** Caroline Twarog, Sarinj Fattah, Joanne Heade, Sam Maher, Elias Fattal, David J. Brayden

**Affiliations:** 1UCD School of Veterinary Medicine and UCD Conway Institute, University College Dublin, Belfield, Dublin 4, Ireland; caroline.twarog@ucdconnect.ie (C.T.); sarinj.fattah@ucd.ie (S.F.); joanne.heade@ucdconnect.ie (J.H.); 2School of Pharmacy, Royal College of Surgeons in Ireland, St. Stephen’s Green, Dublin 2, Ireland; sammaher@rcsi.ie; 3School of Pharmacy, Institut Galien, CNRS, Univ. Paris-Sud, Univ. Paris-Saclay, 92290 Châtenay-Malabry, France; elias.fattal@u-psud.fr

**Keywords:** oral macromolecule delivery, oral peptides, sodium caprate, salcaprozate sodium, epithelial permeability, epithelial transport

## Abstract

Salcaprozate sodium (SNAC) and sodium caprate (C_10_) are two of the most advanced intestinal permeation enhancers (PEs) that have been tested in clinical trials for oral delivery of macromolecules. Their effects on intestinal epithelia were studied for over 30 years, yet there is still debate over their mechanisms of action. C_10_ acts via openings of epithelial tight junctions and/or membrane perturbation, while for decades SNAC was thought to increase passive transcellular permeation across small intestinal epithelia based on increased lipophilicity arising from non-covalent macromolecule complexation. More recently, an additional mechanism for SNAC associated with a pH-elevating, monomer-inducing, and pepsin-inhibiting effect in the stomach for oral delivery of semaglutide was advocated. Comparing the two surfactants, we found equivocal evidence for discrete mechanisms at the level of epithelial interactions in the small intestine, especially at the high doses used in vivo. Evidence that one agent is more efficacious compared to the other is not convincing, with tablets containing these PEs inducing single-digit highly variable increases in oral bioavailability of payloads in human trials, although this may be adequate for potent macromolecules. Regarding safety, SNAC has generally regarded as safe (GRAS) status and is Food and Drug Administration (FDA)-approved as a medical food (Eligen^®^-Vitamin B_12_, Emisphere, Roseland, NJ, USA), whereas C_10_ has a long history of use in man, and has food additive status. Evidence for co-absorption of microorganisms in the presence of either SNAC or C_10_ has not emerged from clinical trials to date, and long-term effects from repeat dosing beyond six months have yet to be assessed. Since there are no obvious scientific reasons to prefer SNAC over C_10_ in orally delivering a poorly permeable macromolecule, then formulation, manufacturing, and commercial considerations are the key drivers in decision-making.

## 1. Introduction

Despite an increasing trend in drug discovery and development in favor of biologics (macromolecules), poor oral availability remains a major impediment to even more widespread application. One group of macromolecules, peptides and proteins, are especially advocated due to excellent specificity, selectivity, safety, and efficacy. Indeed, a combined ~240 were marketed since the 1980s [1]. Of that list, 12% have <60 amino acids, designating approximately 30 peptides from that total [2]. Over 90% of peptides are injectable formulations, with just 4% delivered orally, and even lower percentages delivered via the skin and airway routes [2]. Recent progress was made toward the development of oral formulations for peptides where there are scientific, patient acceptability, and commercial arguments for non-injectable alternatives, especially for those that are used chronically and require frequent dosing (e.g., glucagon-like peptide 1 (GLP-1) analogs) [3]. The oral route offers greater patient compliance and can generate large market sales for molecules working indirectly on the same overall biological target, even if overall efficacy is lower than parenteral options. This is the case for oral dipeptidyl peptidase 4 (DPP-4) small-molecule inhibitors in competition with injectable GLP-1 peptide analogs [4]. Oral administration of peptides is limited by local conditions within the gastro-intestinal (GI) tract except for two relatively low-molecular-weight (LMW) examples designed for systemic delivery: a microemulsion of cyclosporin (Neoral^®^, Novartis, Switzerland) and a conventional solid-dose formulation of desmopressin (Minirin^®^, Ferring, USA) [5]. However, these examples are exceptions based on the atypical macrocycle structures of the two peptides, yielding oral bioavailabilities (BA) of 30–40% for lipophilic cyclosporine in Neoral^®^ and just 0.17% for the highly potent hydrophilic desmopressin, Minirin^®^ [6]. 

## 2. Challenges for Oral Delivery of Macromolecules

Oral administration of hydrophilic macromolecules with a molecular weight (MW) above 1000 Da remains a challenge due to susceptibility to pH and gastric/small intestinal enzymes, as well as low intestinal epithelial membrane permeability. The low permeability results from minimal passive or carrier-mediated transcellular permeation across phospholipid bilayers, as well as restricted paracellular transport via tight junctions. If they were small molecules, peptides would likely be assigned to Class III of the biopharmaceutics classification system (BCS), typically exhibiting high aqueous solubility (but not always) and low intestinal permeability. It is noteworthy that, even in the example of cyclosporine where the fraction absorbed (f_abs_) is high, sensitivity to intestinal cytochrome P450 metabolism and P-glycoprotein efflux reduce the BA [7]; its primary problem is intestinal wall metabolism, not permeability. Other variables also impact the feasibility of oral delivery of macromolecules. If the plasma half-life (t_½_) is too short, it will not be economically viable to administer a peptide candidate in multiple daily oral doses, where safety, efficacy, and variability issues would also arise. Similarly, a large therapeutic index (TI) is important in the context of selecting potent macromolecules as oral candidates, since efficacy and safety need to be addressed at the low and variable BA values that may be achieved even with successful oral formulations.

Investigators attempted to address pre-systemic degradation and poor permeation in the same formulation. A common approach is to combine peptidase inhibitors with absorption-modifying excipients (AMEs) or chemical permeation enhancers (PEs). These are usually formulated in enteric-coated dosage forms [6], although those formulated with salcaprozate sodium (SNAC), the leading candidate PE of the Eligen^®^ technology (Emisphere, NJ, USA), do not seem to require coating [8]. In addition to avoiding degradation by gastric enzymes and low pH, enteric-coated capsules and tablets avoid dilution and premature release of both PE and macromolecule in the stomach. Furthermore, coatings can assist in promoting co-release of both in high concentrations at the same region to maximize intestinal permeability [9], a formulation goal to maximize payload delivery. Incorporation of PEs in conventional oral dosage forms is considered a relatively basic technology approach to address macromolecule permeability [10]. However, the ease with which PEs can be incorporated into delivery systems without the need for sophisticated and costly formulation made them more commercially attractive compared to, for example, nanotechnology [11] and device-based systems [12]. The majority of formulations currently in clinical trials for oral peptides are, therefore, based on PEs, whereas most nanotechnology and device-based systems remain in preclinical research [6]. This scenario may change if PE-based formulations only prove efficacious and commercially viable for exceptions: highly potent, stable, long-t_½_ molecules of relatively low MW, and with a large TI.

## 3. Intestinal Permeation Enhancers

Numerous compounds including surfactants, bile salts, bacterial toxins, chelating agents, and medium-chain fatty acids (MCFA) proved to be effective PEs for poorly permeable molecules in in vitro and in vivo studies [10,13]. A comprehensive analysis of the majority of intestinal PEs from these classes that are used with peptides is available [10]. PEs that increase permeability across Caco-2 monolayers, isolated intestinal tissue mucosae, and in rodent models may also improve oral BA in humans, but this is not guaranteed since such studies are predominantly based on admixtures with payloads, not oral formulations. Furthermore, scale-up of the final formulation, PE dose, dilution, spreading, and release of both PE and payload during transit in the human GI tract, as well as the influence of enzymes, bile salts, and lipids in human intestinal fluids, must all be taken into account when attempting to make oral BA predictions for humans from preclinical studies. There are currently over 50 clinical trials in which PEs were shown to increase oral absorption of poorly permeable molecules, mostly achieved using surfactants [14]. The most widely tested PEs in these trials include Eligen^®^ carriers, MCFAs, acyl carnitines, bile salts, and ethylenediaminetetraacetic acid (EDTA) [15]. The MCFA, sodium caprate (C_10_), and the C_8_ derivative, salcaprozate sodium (SNAC), are of particular interest as they had over 20 years of development in proprietary delivery platforms and have been tested in human trials more than any other PEs. C_10_ was originally developed as the main component of an oral solid-dosage form (GIPET^TM^, Gastro-Intestinal Permeation Enhancement Technology) by Elan Pharma (Dublin, Ireland), and then, following licensing, by Merrion Pharmaceuticals (Dublin, Ireland) for oral peptide delivery, and by Ionis Pharmaceuticals (Carlsbad, CA, USA) for oral delivery of antisense oligonucleotides. SNAC was developed by Emisphere (NJ, USA) as the lead agent of its Eligen^®^ carrier technology. Novo Nordisk (Bagsværd, Denmark) licensed both GIPET^TM^ and Eligen^®^ to assess with their insulin and GLP-1 analogs, ultimately opting to focus on an SNAC tablet formulation with their highly potent, stable, long-t_½_ (160 h) injectable GLP-1 analog, semaglutide, for advanced clinical development, while abandoning GIPET™, along with further attempts to create an oral insulin.

## 4. Introducing C_10_

C_10_ is the sodium salt of capric acid, an aliphatic saturated 10-carbon MCFA (Figure 1A). Fatty acids are ubiquitous nutrients liberated in high quantities during digestion of glycerides in the GI tract. They are also present in low mM concentrations in various nutrient sources, including milk. C_10_ is approved as a food additive in both the United States (US) and European Union (EU) and there are no daily limits on consumption; it was recently concluded that its presence in food should have no impact on human health [16]. C_10_ was a component of an approved rectal suppository of ampicillin (Doktacillin^®^, Meda, Solna, Sweden) [17]. It was since assessed in clinical trials by Merrion Pharma as oral solid-dosage forms (GIPET^TM^) for the delivery of a wide range of poorly permeable actives, including small molecules (e.g., zoledronic acid, alendronate) and macromolecules (insulin, desmopressin, acyline, and antisense oligonucleotides) [18]. C_10_ is a soluble anionic surfactant, sensitive to changes in pH and ionic strength. At pH values 1–3 units below its pKa (~5) in gastric fluid, it exists in the non-ionized, insoluble, and inactive capric acid form. At acidic pH values, the surfactant can reduce surface tension, but does not exhibit detergent action. At pH values 1–3 units above its pKa (i.e., values that typically occur in the small intestine), C_10_ exists in an ionized soluble form with detergent capacity. Like many other efficient detergents, it does not form micelles efficiently owing to repulsion between the charged hydrophilic head groups. The resulting high concentration of free monomeric surfactant enables epithelial plasma membrane interaction and confers a transcellular element to its mode of action. The critical micellar concentration (CMC) value of C_10_, like other ionizable surfactants, varies depending on the medium composition. Micelles form at lower concentrations in higher-ionic-strength buffers because the counter-ions in media interact closely with anionic head groups. Thus, varying the ionic strength alters the free monomeric concentration of C_10_ in the small intestine.

## 5. Introducing SNAC

SNAC is a synthetic *N*-acetylated amino-acid derivative of salicylic acid (Figure 1B). It was discovered as part of a screen to identify carrier-based PEs that could “chaperone” poorly permeable payloads across the intestine [19]. The carrier library of over 1500 compounds was collectively termed Eligen^®^, and it formed the portfolio of Emisphere Technologies. SNAC is the most extensively tested carrier and the only PE approved in an oral formulation designed to improve oral BA, albeit with a small molecule, cyanocobalamin/SNAC [20]. It is important to note that this oral form of vitamin B_12_ was approved under the regulatory pathway for medical foods, which does not have to meet the standards required for drug approvals, although the regulatory requirements for medical foods are still much higher than those of dietary supplements [21]. Emisphere obtained generally recognized as safe (GRAS) status for SNAC, which was a requirement for developing cyanocobalamin/SNAC for the medical food regulatory pathway. Having GRAS status for this PE may have somewhat mitigated some of the perceived risks associated with the oral semaglutide program at Novo Nordisk. In the 1990s, initial focus on SNAC was aimed at developing an oral formulation of the poorly permeable macromolecule, heparin [22]. In other preclinical studies, it also improved intestinal permeation of peptides (salmon calcitonin (sCT) and insulin) [23], along with poorly permeable small molecules (e.g., cromolyn) [24]. SNAC was tested in many formats: taste-masked liquids, tablets, and soft gelatin capsules. Similar to C_10_, SNAC can be blended with the active pharmaceutical ingredient (API) using conventional processes, which makes manufacturing uncoated oral tablet dosage forms economic and relatively easy to scale. 

It remains unclear if the high concentrations of SNAC required to improve small intestinal epithelial permeation relate to membrane perturbation, membrane fluidization, payload solubility changes, or tight junction openings, or whether it is a chaperone system that improves transcellular permeation via hydrophobization of the payload through non-covalent linkages. Of these factors and, unlike C_10_, there is less direct evidence for tight junction involvement in the mechanism of SNAC than the other factors; hence, its common designation is as a transcellular PE. Common features are that C_10_ and SNAC are weak acids that display amphiphilicity and surface activity. However, there is a structural difference between them arising from the greater distribution of hydrophilic functional groups in the salicylamide region of SNAC, as evident from its higher polar surface area (89.5 Å^2^) compared to C_10_ (40.1 Å^2^) [25,26]. It follows that the hydrophobic region of SNAC should be less efficient at inserting into phospholipid membranes than C_10_ [27]. This may be one of the reasons why higher concentrations of SNAC than C_10_ are needed to improve permeation.

## 6. How Do SNAC and C_10_ Alter GI Permeability?

### 6.1. Challenges in Determining Mechanism of Action

PEs can improve permeability via a combination of mechanisms. Such mechanisms include opening tight junctions to increase paracellular permeability, decreasing mucus viscosity, inhibition of epithelial efflux pumps, complexation/hydrophobization of payload, increasing membrane fluidity, and (indirectly) via peptidase inhibition. The various mechanisms of action of C_10_ and SNAC were studied using cell biology and physicochemical techniques including membrane fluorescence, Western blotting, electrophoretic mobility, molecular imaging, and physical analysis. More recent approaches to the study of the interactions between PEs and payloads use surface plasmon resonance [28], as well as accelerated capillary electrophoresis, and isothermal titrated calorimetry (ITC) [29]; however, these techniques are mostly restricted to simple physiological buffers rather than bio-relevant intestinal fluids. In particular, a concern is that intricate mechanisms determined in in vitro assays might not reflect the true mechanism, because the PE concentrations used in vitro are typically lower than the efficacious doses used in vivo. There is, therefore, uncertainty regarding the actual local concentrations of PE and payload at the small intestinal epithelial wall in a particular region due to variability in dissolution, spreading, and dilution in the human GI lumen during transit.

### 6.2. C_10_ Mode of Action

The mode of action of C_10_ was studied at great lengths in a range of delivery models. In summary, concentrations that lead to alteration to permeability coefficients or oral BA are associated with mild mucosal damage and other hallmarks of transcellular perturbation. At low concentrations, increases in permeability of hydrophilic small molecules across Caco-2 monolayers using relatively low concentrations of C_10_ (2.5 mM) can be uncoupled from loss of monolayer integrity, accompanied by reversible reductions in transepithelial electrical potential (TEER) [27,30], indicative of a paracellular mechanism. The higher concentrations required to alter permeability in isolated rat and human intestinal tissue mucosae are associated with transcellular perturbation [31,32]. Mode-of-action studies at higher concentrations (8–13 mM) in Caco-2 monolayers also allude to a paracellular mechanism involving activation of membrane-bound phospholipase C [33,34,35]. The resulting increase in inositol 1,4,5-triphosphate (IP3) leads to an increase in intracellular calcium (Ca^2+^), which in turn activates calmodulin and myosin light-chain kinase (MLCK). This event triggers the contraction of the peri-junctional actomyosin ring (PAMR) [36], permitting increased tight junction (TJ) permeability. 

Nonetheless, these studies involving pharmacological inhibitors are not definitive proof of a discrete paracellular effect. Impedance spectroscopy in HT29/B6 human intestinal monolayers also supports the dataset showing that C_10_ acts via a paracellular mechanism; this was associated with removal or redistribution of the TJ proteins claudin 5 and tricellulin [37]. However, in the absence of data supporting the absence of transcellular perturbation (a dye uptake assay), the data from this study do not provide conclusive evidence for a paracellular mode of action either. It is impossible to ignore the evidence from a wide range of studies that C_10_ also disrupts cell membranes at 8–13 mM. C_10_ also caused Caco-2 cell leakage of intracellular ATP from Caco-2 cells [33], a likely consequence of plasma membrane perturbation. One interpretation is that cells respond to the initial membrane perturbation challenge by C_10_ with compensating intracellular signaling processes involved in mucosal repair, beginning with disbandment of TJs, and concluded by epithelial resealing [38].

Some of the strongest evidence in favor of a mechanism driven primarily by perturbation, however, comes from high content image analysis in live Caco-2 cells. C_10_ (2.5 mM) increased intracellular Ca^2+^ in Caco-2 cells prior to the plasma membrane permeability changes detected at 8.5 mM [27]. C_10_ (8.5–13 mM) altered both plasma and mitochondrial membrane integrity, indicative of perturbation; importantly, these were the minimum concentrations needed to induce a permeability increase. The elucidation of the primary mode of action as membrane perturbation was clarified by the capacity of C_10_ to preferentially displace claudins 4 and 5 from lipid rafts in MDCK cells, consistent with surfactant properties [39]. Other evidence comes from a recent surfactant screen using isolated rat intestinal mucosae in Ussing chambers, where C_10_ caused a concentration-dependent increase in epithelial histology damage [40]. Given the close association between permeation enhancement and mucosal perturbation in tissue and animal models, it was, therefore, not surprising that the cyto-protectant prostaglandin analog, misoprostol, prevented both the C_10_-induced increase in flux of hydrophilic markers across Caco-2 monolayers and cell damage [41]. From these arguments, it is likely that the high concentrations of C_10_ used in tablets also cause a degree of mild reversible mucosal perturbation in vivo, not unlike that seen with aspirin, alcohol, and spicy foods [42]. In a study of a human rectal formulation of ampicillin with C_10_, there was evidence of mild and reversible mucosal perturbation [17], although the data were confounded by the hyper-osmolarity of the formulation. While it is not possible to conclude that mucosal perturbation of the relatively static rectal mucosal compartment extrapolates directly to the small intestine where transit is relatively fast, it is likely that C_10_ causes mild and reversible regional perturbation within a short period at the high concentrations exposed to the small intestinal epithelium prior to its almost complete absorption within minutes.

There is a lack of understanding of the physicochemical aspects of how C_10_ interacts with mixed micelles in the small intestine in the fasted and fed states. Above its CMC of 25 mM in physiological buffer [27], C_10_ forms micelles and there is a distinct ratio of monomer to micellar-bound material, as highlighted in a recent study on alkyl maltosides [43]. In simulated intestinal buffers, it is uncertain whether the payload is incorporated into or adsorbs onto colloidal structures (e.g., mixed micelles, vesicles, lipid droplets), or whether it admixes with the C_10_ monomer. There is resulting confusion over which format the payload permeates. Thus, while one envisages a paracellular permeation route for polar macromolecules due to hydrophilicity, a transcellular pathway may also be available if C_10_-entrapped vesicles and mixed micelles adsorb payload. Figure 2 is a composite of the possible multiple effects of C_10_ on intestinal epithelia. 

### 6.3. SNAC Mode of Action

A different mode of action to that of C_10_ was proposed to explain how SNAC improves intestinal permeability. In the 1990s, Emisphere scientists proposed that SNAC improves passive transcellular permeation via hydrophobization. The hypothesis was that dipole–dipole non-covalent interaction between the carrier and structural moieties of the payload caused a conformational change in the latter, leading to exposure of hydrophobic regions that favor transcellular permeation. The interaction between SNAC and heparin [22,44] and with insulin [45] was, therefore, thought to be based on increased lipophilicity through hydrogen bonding and/or hydrophobic interactions, permitting dissolution of the complex in lipid bilayers. In support of this hypothesis, SNAC at a concentration of 17 mg/mL improved the permeation of insulin, but not that of radiolabeled mannitol across Caco-2 monolayers, suggesting that the effect was neither related to opening tight junctions nor to a decline in barrier integrity, and this interpretation was supported by confocal microscopy [45]. Higher concentrations of SNAC (50 mg/mL) that were more reflective of concentrations used in in vivo studies, however, caused complete loss of TEER and a 36-fold increase in [^3^H]-mannitol permeability in Caco-2 monolayers [44], data that do not permit definitive conclusions to be made regarding mechanism since such high concentrations compromised the Caco-2 model. In isolated rat jejunal mucosae mounted in Ussing chambers, SNAC (33–66 mM) boosted the flux of a polar marker molecule, 6-carboxy-fluorescein (6-CF), but not that of [^3^H]-mannitol across the epithelium and without reducing TEER values [46]. The authors argued that SNAC was indeed exploiting a transcellular pathway and not tight junctions to allow permeation of the hydrophilic polar ionized molecule (6-CF) and, somewhat controversially, they suggested that SNAC was reducing the charge on CF, thereby improving the capacity to partition in the epithelium. Similar to the Caco-2 study [44], when SNAC was added to jejunal mucosae at a concentration of 165 mM, TEER dropped and the permeability coefficient (Papp) of [^3^H]-mannitol was increased [46], denoting a compromising event. Other Caco-2 studies also support a transcellular mechanism; Malkov et al. [47] detected intracellular signal increases ascribed to fluorescently labeled heparin and in the presence of SNAC, whereas immunohistochemistry data indicated that there were no changes of F-actin or the actinomycin ring during heparin flux. Ding et al. [48] used ITC and Fourier-transform infrared (FTIR) spectroscopy to study the interaction between cromolyn and SNAC, and concluded that the aromatic ring of SNAC inserted between those of cromolyn via its 2-hydroxybenzamide motif, leading to an increase in hydrophobicity of the complex and a reduction in cromolyn hydration. Lactate hydrogenase release (LDH) measurements indicated that the increased cromolyn fluxes across Caco-2 monolayers in the presence of SNAC were not associated with cell damage. 

Despite the elegance of the purported complex mechanism (Figure 3) of how SNAC might increase transcellular flux of multiple payloads, the hypothesis is problematic in several respects. If Eligen^®^ carriers acted solely using dipole–dipole interactions via hydrophobization (and not an electrostatic interaction), it would be difficult to envisage a significant increase in passive permeation since the retention of ionized functional groups would impede passive movement across phospholipid bilayers. 

SNAC forms a conjugate base at the pH of the small intestinal lumen, so it can undergo complexation via hydrophobic ion pairing (HIP) with the conjugate acid of basic amino-acid side chains in macromolecules. However, HIP cannot fully account for Eligen^®^-mediated hydrophobization of anionic payloads including heparin and cromolyn [24]. An alternative interpretation arises from another SNAC study with cromolyn; SNAC increased Caco-2 epithelial cell membrane fluidity as measured by fluorescence anisotropy, consistent with a surfactant-induced membrane perturbation effect, whereas in this study there was no increase in cromolyn’s lipophilicity [50]. Still, the overall contribution of transcellular perturbation to the increased flux is not clear since the presence of hydrophilic functional groups in the salicylamide region of SNAC gives rise to inefficient micelle formation (CMC: 56 mM in phosphate-buffered saline (PBS)) [50], and this will not favor membrane insertion. There are a number of other anomalies concerning the original chaperone transcellular mechanism proposed for SNAC (reviewed in Reference [49]). Firstly, since the structure of SNAC comprises MCFA and salicylic-acid moieties, non-specific detergent/surfactant effects on the epithelium should be expected. Secondly, the thermodynamic considerations with respect to the non-covalent linkage between SNAC and payload during epithelial flux are yet to be addressed. Furthermore, there were no calculations on the affinity of SNAC to payloads, except to assert that affinity was weak, now confirmed for exenatide [29]. Thirdly, a transcellular mechanism should account for epithelial endocytosis uptake pathways (e.g., via clathrin- or caveolae-mediated pathways or macropinocytosis), where a template to follow is in place for the other group of transcellular permeability-enhancing agents, the cell-penetrating peptides [51]. Finally, caution must be exercised in making definitive conclusions on mechanism using some of these assays; epithelial TEER values yield information on monolayer integrity, but reveal no direct information about tight junctions. Secondly, the absence of changes in tight-junction-associated antibody imaging for associated proteins is not definitive. Thirdly, fluorescently labeled payloads need to be assessed for their capacity to remain intact during flux. Finally, LDH release from monolayers following exposure to SNAC would not be regarded as a particularly sensitive assay, especially when increased membrane fluidization was observed. In sum, evidence from Caco-2 studies is not yet convincing enough to solely ascribe an exclusive transcellular mechanism for SNAC; moreover, there is little data to directly support the chaperone hypothesis. On the other hand, there are some differences between the mechanism of action of SNAC and those of PEs with mechanisms associated with tight-junction openings (e.g., EDTA).

Recently, Novo Nordisk offered a new mechanism of action for SNAC in its non-enteric coated tablet of the GLP-1 analog, semaglutide (t_½_ = 160 h). Using a ligated dog model, they found that systemic delivery was achieved solely from stomach administration of the tablet [52]. The theory is that SNAC forms a complex around the semaglutide in the stomach and causes a transient increase in local pH around the molecule. It is claimed that semaglutide is protected against pepsin by SNAC and that solubility is increased, resulting in an increased concentration-dependent flux of semaglutide across the gastric mucosa, using a transcellular mechanism as the tablet comes in intimate contact with the epithelium. By shifting the emphasis toward elevation in stomach pH away from conformational changes and increased lipophilicity, the theory takes the focus away from membrane perturbation. Moreover, the authors argue that this mechanism is highly specific for semaglutide, in that similar studies with admixtures of SNAC and liraglutide led to no flux increase across in vitro gastric epithelial models [52]. Part of the role of SNAC seemed to be to convert semaglutide to a more permeable monomeric form and it seems to perform this better when formulated in a stomach-specific tablet. Is this payload-specific and region-specific theory entirely compatible with the previous data from small intestinal studies in which SNAC was paired with many payloads of differing structures? With new cell-imaging tools available, along with advanced biophysical methods to decipher the interaction between SNAC and payloads, it is likely that much of the discrepancy surrounding the mechanism of SNAC will be resolved. Figure 4 is a composite of the possible effects that SNAC has in the stomach when formulated with semaglutide.

In sum, the in vitro studies on the mechanism of action of the two agents on cultured intestinal epithelia suggest some common surfactant-based features; however, in contrast to SNAC, there is direct evidence for tight-junction openings induced by C_10_. The culture models are sub-optimal, however, as they do not discriminate permeation enhancement from perturbation very well, nor do they predict in vivo consequences; the models have difficulty in tolerating both high concentrations of PEs and simulated intestinal fluids. 

## 7. C_10_ and SNAC: Pharmacokinetics and Efficacy in Clinical Trials

### 7.1. C_10_

GIPET™ was advanced to clinical testing as enteric-coated tablets containing C_10_ with both peptide and small-molecule payloads [18,42]. Human studies using radiolabeled polyethylene glyol (PEG) revealed that the permeating enhancement effects of GIPET™ were transient and reversible in <1 h [18]. GIPET™ was tested in a range of doses with several poorly absorbed molecules in a total of 16 Phase I studies comprising over 300 subjects [18]. Overall, while oral BA values of >5% were cited for some molecules, the most notable feature was the massive intra-subject variability across all studies, constituting an issue for safety and efficacy. 

Pharmacokinetics (PK) analysis of human trials for GIPET™ formulations with low-molecular-weight heparin (LMWH) and desmopressin was described [18]. LMWH–GIPET™ was formulated in tablets containing either 45,000 or 90,000 IU of LMWH at two dose levels of C_10_. Oral BA was calculated relative to the standard sub-cutaneous (s.c.) dose of 3200 IU following administration to 14–16 subjects. Relative oral BA of 3.9–7.6% was achieved [18]. With a high dose of LMWH combined with a high dose of C_10_, increased levels of an anti-clotting biomarker were seen in all subjects; the responses were sustained and had a similar time course to the s.c. route. This particular formulation was not progressed clinically (Table 1). When desmopressin was formulated with GIPET™ and administered orally to 18 human subjects, a bioavailability of 2.4% relative to the s.c. route was detected [18], an improvement over the typical 0.2% value for Minirin^®^ tablets. Again, this formulation was not progressed further. The GIPET™ technology was well tolerated even when repeatedly administered in these small Phase I studies [18]. Other clinical trial examples include the gonadotropin-releasing hormone (GnRH) antagonist decapeptide, acyline. In a Phase I study of oral acyline, serum luteinizing hormone (LH), follicle-stimulating hormone (FSH), and testosterone were suppressed within 12 h at the 10-, 20-, and 40-mg single doses tested. However, sustained serum levels of acyline could not be detected, and there was no pharmacokinetic–pharmacodynamic (PK–PD) relationship [53]. The GIPET™ technology was also used to orally deliver the bisphosphonate, zoledronic acid. The rationale was that an oral tablet (Orazol™) administered weekly by patients could compete with a monthly infusion of Zometa^®^ in a hospital setting for cancer patients with bone metastasis. In a Phase I study, urinary excretion of unchanged zoledronic acid suggested equivalent delivery via both routes [54]. The licensing of GIPET™ to Novo Nordisk led to Phase I trials with respect to a proprietary insulin, NN1953, and a GLP-1 analog; however, the resulting data were never published and, ultimately, Merrion’s remaining intellectual property (IP) assets were sold to Novo Nordisk, before being liquidated in 2016. Novo Nordisk in turn decided to move away from developing oral insulin to concentrate on its oral GLP-1 analog program. Nonetheless, an important Phase II trial from that time assessing a once-daily long-acting basal insulin (I338) with a t½ of 70 h in a GIPET™ formulation was published in 2019 by Novo Nordisk [55]. In this study, a relative oral F versus the long-acting s.c.-administered insulin glargine (Lantus^®^, Sanofi, Paris) of 1.5–2.0% was achieved without evidence of toxicity over eight weeks. Although similar plasma glucose reduction was achieved by the oral GIPET™-based formulation to the s.c. insulin, the rationale for discontinuation was that the dose of I338 was not commercially viable. The GIPET™ journey with oral peptides and poorly permeable small molecules, therefore, ended without generating a product in 2015. As a post-script, a Phase I study was published in 2018 by Biocon (India) in which a C_10_-based formulation of their alkylated PEGylated fast-acting meal-time insulin (IN-105; Insulin Tregopil) was shown to have no effect on the PK of oral metformin in fasted conditions and it was well tolerated [56]. Therefore, C_10_ continues to be used in oral peptide formulations both in clinical trials and as a comparator for other PEs in preclinical studies.

The other arm of the original Elan licensing of C_10_-based matrix tablets in the late 1990s continued in parallel with respect to antisense oligonucleotides. The gene medicine specialty Pharma, Ionis Pharmaceuticals (Carlsbad, CA, USA) (formerly Isis Pharma) developed a number of oral antisense oligonucleotide formulations containing C_10_ for clinical testing against RNA targets. One candidate that progressed to Phase I was ISIS 104838, a tumor necrosis factor (TNF)-α inhibitor. Oral administration of a C_10_-based tablet to dogs resulted in average absolute oral BA of 1.4% [57]. Tissue histology of the small intestine and large intestine of the dogs indicated no changes following once-daily dosing of tablets containing ~1 g of C_10_ over seven consecutive days. A subsequent Phase I trial examined ISIS 104838 (100 or 140 mg) formulated with C_10_ (660 mg) in immediate-release mini-tablets packaged in enteric-coated gelatin capsules, with or without a second mini-tablet containing only C_10_. The second group of mini-tablets was coated with different layers of Eudragit^®^ RS30D to allow for subsequent further release of the C_10_ following erosion of the first tablet containing ISIS 104838 [58]. The goal was to create a greater window for absorption by prolonging the time C_10_ was in contact with the epithelium, given that it is rapidly absorbed with a T_max_ of 7 min. All formulations together yielded an average oral BA of 9.5% relative to s.c. injection, with the formulation designed for additional immediate release of C_10_ giving a value of 12%; however, the intra-subject variability ranged from 2–28% [58]. In 2017, Ionis advanced an oral antisense molecule IONIS-JBI1-2.5Rx, aimed at an RNA target associated with a GI autoimmune disorder, to Phase I trials in collaboration with Janssen (Beerse, Belgium) [59]; however, it is unlikely that the formulation contains C_10_ as it is designed for local colonic delivery. Table 1 summarizes the clinical data reported for a range of poorly permeable molecules with C_10_. Of these, there are only four peer-reviewed original research papers for GIPET™ tablets. 

### 7.2. SNAC

SNAC was in a succession of clinical trials in oral formulations with poorly permeable actives since the late 1990s, culminating with approval for cyanocobalamin as a medical food for vitamin B_12_-deficient anemic subjects in 2014 [20,21]. The initial clinical trials were carried out using unfractionated heparin in 1998 [61]. In the first Phase I study, 2.25 g of taste-masked SNAC was combined with 30,000–150,000 IU of heparin and administered to subjects via gavage; the formulation achieved increases in outputs associated with anti-coagulation efficacy: activated partial thromboplastin time (aPTT) and production of anti-factor Xa. This led to subsequent Phase I and II trials with taste-masked 10–15-mL liquid formulations in patients undergoing total hip replacements; oral heparin was dosed at either 60,000 or 90,000 IU with 1.5 or 2.25 g of SNAC respectively, and results were compared to s.c. administration of 5000 IU of heparin [62]. The oral dosing regimen comprised 12–16 doses over a four-day period after surgery. Data from the second Phase I study showed that the oral heparin liquid formulation induced anti-factor Xa activity similar to s.c. heparin. In the Phase II trial, major bleeding events were similar between oral and s.c. heparin groups, thereby offering encouragement on the safety front. In 2002, the oral liquid heparin formulation ultimately missed its primary efficacy end-point in a Phase III trial (PROTECT) comparing oral heparin (either 60,000 IU/1.5 g SNAC or 90,000 IU/2.25 g SNAC three times a day) to s.c. LMWH (enoxaparin) over a 30-day period with assessment for deep vein thrombosis as the read-out. The study comprised over 2000 patients undergoing elective hip replacement, a study that was associated with poor compliance due to the bitter taste of the solution [49]. Direct leveraging from a taste-masked drink to a solid dosage form was not possible, due to the high quantities of SNAC and heparin. Subsequently, a new Phase I PK–PD study was eventually carried out in 2007 in 16 subjects receiving a 75,000 IU heparin/500 mg SNAC total dose in soft gel capsules [63]. It confirmed the effect on aPTT and the orally delivered heparin had a C_max_ of 58 min. Ultimately, a solid dose formulation of oral heparin/SNAC never reached Phase III and was abandoned, perhaps in part because of the advantages of LMWH over unfractionated heparin, as well as the advent of alternative oral anti-thrombotics. 

In relation to other payloads and Eligen^®^ carriers, a caprylic-acid derivative, possibly SNAC, was also formulated with sCT, and a Phase I study in eight volunteers was published in 2002, with a comprehensive analysis of PK [64]. This benchmark study described tablets of 0.4 mg of sCT with 225 mg of SNAC that were dosed singly, in duplicate, or in triplicate to present individual doses of 0.4, 0.8, and 1.2 mg of sCT. The resulting absolute oral BA values versus an intravenous (i.v.) dose of 10 µg ranged from 0.5–1.4%. An Eligen^®^ formulation of insulin was also assessed in a 2010 trial in 14 Type 2 diabetics (T2D) where the carrier was monosodium *N*-(4-chlorosalicyloyl)-4-aminobutyrate (4-CNAB). An oral BA of 7 ± 4% was achieved from a 300-IU dose (with 200–400 mg of 4-CNAB) versus an s.c. dose of 15 IU in fasting subjects [65]. Such large variability would not be acceptable for this low-therapeutic-index drug. A third related Eligen^®^ carrier, 8-(*N*-2-hydroxy-5-chloro-benzoyl)-amino-caprylic acid (5-CNAC), was also evaluated by Novartis (Switzerland) and Nordic Biosciences (Herlev, Denmark) in three large Phase III trials for oral sCT: two for osteoarthritis [66] (NCT00486434 and NCT00704847) and one for osteoporosis [67] (NCT00525798). The dosing to several thousand patients across the three trials comprised one tablet (0.8 mg sCT with 200 mg of 5-CNAC) in a tablet administered twice a day with 50 mL of water approximately 30 min ahead of meals. These studies lasted 24 months for the osteoarthritis trials and 36 months for the osteoporosis trial. Although these trials missed their primary efficacy end-points, interesting assessments concerning dosing formats and regimes were published with conclusions that may have relevance for SNAC trials [68]. Differences in sCT absorption and effects on bone biomarkers occurred depending on the volume of water, proximity to a meal, and the time of day (reflecting circadian rhythms in bone turnover). Table 2 summarizes the clinical trial performance of oral SNAC and related Emisphere carrier formulations across a range of poorly permeable molecules.

Recent focus, however, shifted entirely to the clinical development of the oral semaglutide/SNAC tablet by Novo Nordisk. Once-daily oral semaglutide with 300 mg of SNAC resulted in improved glycemic control and greater reductions in body weight than placebo in a 26-week Phase II dose-escalation study in doses ranging from 2.5–40 mg of semaglutide per day in over 600 patients with T2D [69]. Daily oral administration of semaglutide (20 mg and 40 mg) with SNAC lowered glycated hemoglobin (HbA1c) by over 1.4% and these data were comparable with that seen with weekly s.c. administration of semaglutide (1 mg). For oral semaglutide with SNAC, the oral BA is likely to be ~1%, although the focus of the publications was on the PD effect and biomarkers. Clues come from Beagle dog studies where tablets containing 300 mg of SNAC with 5–20 mg of semaglutide gave oral BA values of 1.22 ± 0.25% following oral administration [52,70]. Novo Nordisk completed ten Phase IIIa PIONEER trials in 2018. Top-line data from PIONEER 1 achieved significance with respect to a reduction in HbA1c of 1.5% with a 14-mg semaglutide dose in T2D, along with evidence of some weight loss [71]. Recent trials also revealed that renal impairment did not affect PK parameters of 5 mg and 10 mg of semaglutide formulated with 300 mg of SNAC over a short time frame in diabetics [72]. This design was repeated in diabetic patients with hepatic impairment with the same outcome in that PK values were not altered and, therefore, no dose adjustment was needed in these patients [73]. The question of the impact of omeprazole on PK was also assessed, as elevation of bulk stomach pH might have confounded the purported mechanism of SNAC. Using a 5-mg semaglutide dose in patients taking 40 mg of omeprazole over a 10-day period with a follow-up period out to 21 days, overall PK values for both semaglutide and SNAC were unchanged, leading to a conclusion that dose adjustment would be also unnecessary in patients on concomitant omeprazole [74]. The implications of these findings would support the determination that the pH increase created by SNAC in the stomach must be at the semaglutide tablet surface [52] and does not impact bulk stomach pH; otherwise, a large effect of omeprazole on PK would have been expected. The PIONEER 6 Phase III study enrolled diabetics with cardiovascular disease in order to exam oral semaglutide PK and PD in this cohort to see if the daily 14-mg formulation increases cardiovascular risk [75]. Selected oral semaglutide clinical data are summarized in Table 3. It will be interesting to examine patient compliance with the current rather inconvenient dosing regime, especially in post-marketing studies if oral semaglutide is approved, since the daily tablet must be taken at least 30 min before meals in the morning in order to avoid food interference with formulation performance.

## 8. Safety of SNAC and C_10_ in Preclinical and Clinical Studies 

Although C_10_ was previously marketed in a rectal product, this has limited relevance to the safety of an orally delivered tablet formulation. The approval of SNAC in an oral vitamin-B_12_ medical food product, though encouraging, is also only partially informative. Nonetheless, the clinical trial experience with both PEs in hundreds of subjects over more than 20 years suggest that only very low numbers of subjects experienced side-effects that caused drop out from trials, and the majority of reports were related to mild GI effects including nausea and diarrhea for GIPET™ [18,42] and SNAC [69]. Reversibility studies performed with C_10_ in humans using the lactulose:mannitol urinary excretion ratio (LMER) assay showed that, following intra-jejunal administration to human subjects, the enhancer only increased permeability in a 20-min window [42]. It seems that dilution, spreading, and rapid intestinal absorption of both C_10_ and SNAC prevents prolonged exposure in vivo. There is no direct evidence, even from studies with a duration as long as six months, that stomach or duodenal ulcers are caused by these PEs, nor that pathogens can gain entry across a compromised intestinal epithelium. Still, post-marketing surveillance will provide more safety data in the context of daily administration over several years, at least in the case of SNAC with both vitamin B_12_ and semaglutide in the event of Food and Drug Administration (FDA) approval. Additional safety experience for SNAC will be ascertained from extensive Phase III oral semaglutide trials, which will fully report in 2019. 

In terms of preclinical safety data, the experience for both molecules is extensive. Numerous studies reveal little toxicity of high doses of C_10_ in rats, dogs, and pigs following oral administration alone and in combination with payloads [18,42,76]. In a study of the acute effects of a C_10_-based dosage form (Orasense™) in Beagles, Raoof et al. provided evidence of the safety of oral hydroxypropyl methyl cellulose-coated C_10_/antisense tablets [57]. Several hundred milligrams of C_10_ were used in each tablet and dogs received treatment three times per day for seven days. Clinical chemistry and blood biochemistry parameters were normal; the dogs tolerated the formulation and there was normal weight gain. Canine intestinal issues were also adjudged normal following macroscopic examination post mortem. Five separate canine daily tolerance studies revealed encouraging safety data for selected components of the GIPET I and II technology [42]. Similar studies were also carried out intra-intestinal catheterized pigs where C_10_ was formulated with antisense oligonucleotides at doses up to 100 mg/kg of the MCFA; it was well tolerated following multiple doses and with little evidence of intestinal epithelial damage post mortem [77]. 

For SNAC, Riley et al. carried out a sub-chronic oral toxicity test of SNAC in rats and found a no observed adverse effect (NOAEL) level of 1 g/kg/day in rats for up to 13 weeks; it was only a massive dose of 2 g/kg/day that eventually caused significant mortality [78]. It was also examined for gestational toxicity in pregnant rats at oral doses up to 1 g/kg/day where slight weight loss was seen; there was no effect on growth of pups, but some evidence of a small increase in the still-birth rate was noted [79]. Some GI effects including emesis and diarrhea were observed in studies involving monkeys at a dose of ≥1.8 g/kg/day [80]. SNAC ultimately achieved provisional GRAS status as a food additive. The safety data from clinical and preclinical studies, therefore, raised no red flags for either agent in oral dosage forms at high concentrations with a wide range of actives tested to date. An important safety consideration is the high inter-subject variability typically associated with the low oral bioavailability values for all payloads tested with both PEs to date; formulation with these PEs will, therefore, not be suitable for molecules with low therapeutic indices.

It is clear that surfactant-based PEs cause a mild degree of reversible perturbation of the intestinal mucosa. The small intestinal epithelium is entirely renewed every 72 h [81]; there is a high rate of cellular turnover and the intestine has a high capacity to replace cells, due to migrating stem cells from the intestinal crypt. There is also a reserve of stem cells that are dormant until epithelial injury occurs, at which point they are recruited to assist with restoration [82]. The capacity of the intestinal mucosa to repair is also associated with secretion of mucus, prostaglandins, and bicarbonate [80]. For a comprehensive review of repair and restoration of the intestinal barrier, see Blikslager et al. [38]. The capacity for epithelial repair following C_10_ exposure was investigated in rat jejunal instillations, where full restitution was seen within 60 min of exposure [83]. These data were similar to that seen in rat models with other PEs including bile salts [84] and sodium dodecyl sulfate (SDS) [85]. Since C_10_ and SNAC are rapidly and completely absorbed, one interpretation is that, following local and transient mucosal perturbation leading to a transient increase in permeability, the epithelium recovers due to gradual dilution of the PE.

Another concern over routine use of PEs is based on their potential capacity to promote microbiome changes and absorption of microorganisms, antigens, and toxins leading to local inflammation, autoimmune disease, and sepsis [86]. Surfactants may impair the protective mucus layer, facilitating the diffusion of luminal bacteria to the intestinal epithelium and ultimately disturbing the host microbiota [87]. In a recent study which generated much debate, evidence was provided that the approved excipient emulsifiers, polysorbate-80 and carboxymethyl cellulose, disturb microbiota composition and induce obesity in mice [88]. Whether these data have any true significance for humans is not known at this point, but it is clear that intestinal microbiome research is going to become more relevant in toxicology profiling of oral formulations. A second concern that is continually raised is the potential increase in permeability of bystander molecules arising from tissue damage induced by PEs [89,90]. Taking into account the precise conditions required for permeation enhancement (high concentrations of payload and PE contemporaneously at the intestinal epithelium), as well as the marked difference in the MW of candidate payloads (<10 kDa) compared to that of typical bacteria, viruses, and bacterial lipopolysaccharide (LPS) (>100 kDa), this concern may be overstated. Nonetheless, clinical pharmacology data from binge-drinking human subjects suggests that alcohol can permit absorption of endotoxins and can promote elevation of type 1 cytokines in plasma, akin to a low-grade infection [91]; thus, together with the study of relevant microbiome changes, more research is needed to filter the true toxicological risks of orally delivered PEs following chronic exposure. 

Several studies describe an anti-microbial effect of C_10_ at high concentrations. Cox et al. [92] demonstrated the bactericidal property of C_10_ against *Salmonella typhimurium*, and it also prevented its attachment to intestinal epithelia. Moreover, there was no evidence from the same study that C_10_ promoted permeation of this gut pathogen across isolated rat intestinal mucosa. At low mM concentrations, C_10_ is also bactericidal against *Helicobacter pylori* [93]. In an in vivo study with chickens, incorporation of C_10_ in feed at a level of 3 g/kg protected them from colonization by *Salmonella enterica* [94]. Finally, capric acid has antifungal activities on *Microsporium gypsum* mycelia and spores in vitro [95]. These data are consistent with the well-known anti-microbial actions of MCFAs [96]. Although, to our knowledge, similar data are not reported for SNAC, it would be surprising if, upon examining its structure, it did not have similar anti-microbial actions. 

## 9. Conclusions

In comparing the safety and efficacy of C_10_ and SNAC as PEs in preclinical and clinical studies, examination of 20–30 years of literature would suggest that several of the key parameters are similar. Both can permit oral bioavailability of a range of macromolecular payloads by <5%, with mean values closer to ~1%. The SNAC clinical PK data with semaglutide seem to be on a par with previous performance, for example, with sCT; however, it is due to its formulation with a potent peptide with a long t_½_ and high TI that is of particular interest. It is the long t_½_ that can compensate for large intra-subject variability [52]. Aspects that tip the balance to SNAC compared to C_10_ include the following: broader clinical experience and an approved vitamin B_12_ product, more extensive toxicology studies and GRAS status, and the lack of requirement for protection against stomach acid. There is still controversy over the mechanism of action of SNAC; techniques including surface plasmon resonance and ITC are now providing data that suggest that the non-covalent interaction of peptides with either SNAC or C_10_ is low affinity and quite similar for each agent. Moreover, while the literature seems to offer a consensus that C_10_ (at low concentrations) acts on tight junctions via intracellular events, and at high concentrations via transcellular perturbation arising from its surfactant effect, there is not the same consensus concerning the mechanism of action of SNAC. The 1990s theory of an exclusive transcellular action arising from increased lipophilicity of a non-covalent complex between SNAC and the payload was not convincing. The new mechanism suggested for SNAC arising from ligated dog studies argues for a local increase in stomach pH around semaglutide, a mechanism that appears to be specific for this molecule. Finally, the main argument advanced for oral peptide delivery is improved convenience over needles leading to better compliance. Patients will, however, be required to wait 30 min before eating and drinking after taking tablets of semaglutide/SNAC each morning; thus, patients will ultimately decide if this is an inconvenience preferable to a once-a-week injection. 

## Figures and Tables

**Figure 1 pharmaceutics-11-00078-f001:**
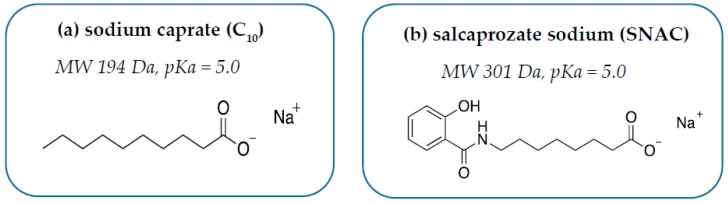
Structures of (**a**) sodium caprate (C_10_) and (**b**) salcaprozate sodium (SNAC).

**Figure 2 pharmaceutics-11-00078-f002:**
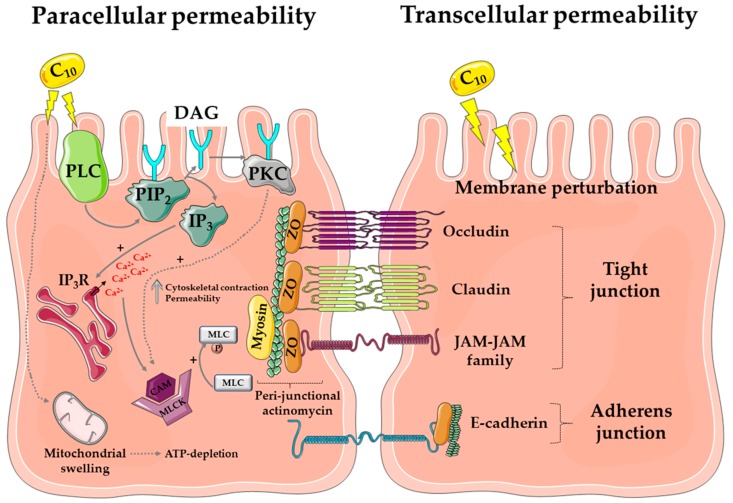
Mode of action of C_10_. The diagram represents the proposed mechanism of action of C_10_ via paracellular flux (**left**) and transcellular perturbation (**right**) to induce drug permeability across the intestinal mucosa. Abbreviations: PLC: phospholipase C; PIP_2_: phosphatidylinositol 4,5-bisphosphate; DAG: di-acyl glycerol; PKC: protein kinase C; IP_3_R: inositol triphosphate receptor; MLC: myosin light chain, CAM: calmodulin; ZO: zonula occludens; JAM: junctional adhesion molecule. Image created using a template from Servier Medical Art under a Creative Commons Attribution License.

**Figure 3 pharmaceutics-11-00078-f003:**
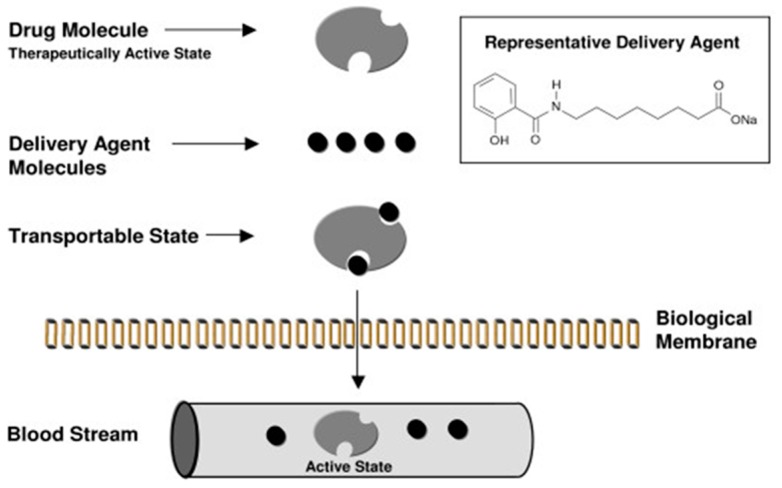
Schematic of Eligen^®^ drug-delivery technology mechanism in intestinal epithelia, as advocated by Emisphere scientists in 2006. “Carrier molecule (delivery agent) associates with drug molecule to create a transportable complex (lipophilic). Because of the weak association, carrier and drug dissociate by simple dilution on entering the blood circulation.” Reproduced from Reference [49] under the terms of the Creative Commons Attribution License.

**Figure 4 pharmaceutics-11-00078-f004:**
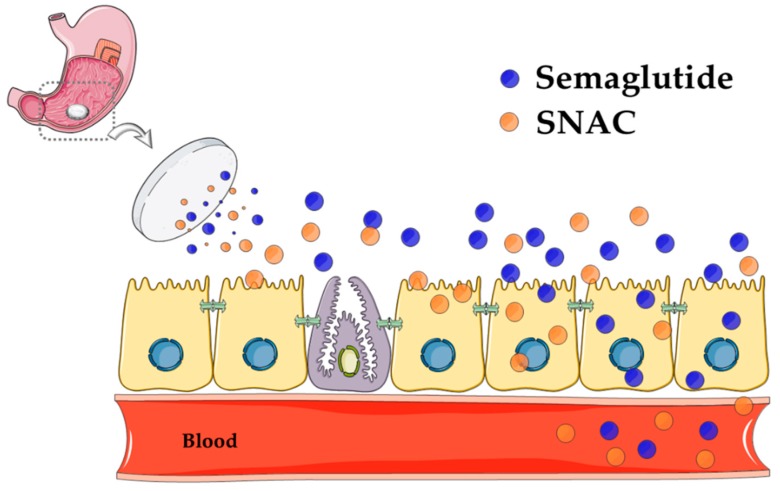
Theory of oral semaglutide absorption, as advocated by Novo Nordisk. Modified from Reference [52]. The diagram represents the proposed mechanism of action of SNAC in inducing transcellular flux of semaglutide across the gastric epithelium of the stomach. The optimum once-daily tablet consists of 14 mg of semaglutide co-formulated with 300 mg of SNAC. After digestion, the tablet erodes rapidly in the stomach, resulting in the release of a highly concentrated amount of SNAC that neutralizes the pH of gastric fluid in the immediate vicinity of the tablet to inactivate pepsin. SNAC is thought to induce semaglutide monomer production and increase gastric epithelial membrane fluidity, but without affecting tight junctions, thereby allowing transcellular passage of semaglutide into systemic circulation. The complex may dissociate at some point in the flux process (Figure 3) due to weak association, but direct evidence for this is scant. Black circles = semaglutide; white circles = SNAC. Image made using a template from Servier Medical Art under a Creative Commons Attribution License.

**Table 1 pharmaceutics-11-00078-t001:** Summary of data from selected studies in humans reported for a range of poorly permeable molecules formulated with sodium caprate (C_10_).

Description	Treatment	Outcome	Reference
Ampicillin with C_10_ in healthy subjects (*n* = 12).	Rectal suppository containing 250 mg of ampicillin and 25 mg of C_10_.	C_max_ increased 2.6-fold compared to ampicillin alone and BA increased 1.8-fold. Some local tissue damage not ascribed to C_10_.	[17]
Phenoxymethylpenicillin, antipyrine with C_10_ in healthy subjects (*n* = 6).	Rectal perfusion containing 2 g of phenoxymethylpenicillin, 8 mg of antipyrine, and 0.7 g of C_10_. Two treatments (T), T1: pH 6 and T2: pH 7.4. Each subject received control (no C_10_) and treatment.	C_10_ was ineffective at increasing permeability across rectal epithelium.	[60]
GIPET™: oral acyline in healthy subjects (*n* = 8).	3 oral tablet doses of acyline: 10, 20, and 40 mg. Subjects received all doses, 1 week apart, under fasting conditions.	Significant reduction in LH, FSH, and testosterone. No serious treatment related adverse effects.	[53]
GIPET™: oral zoledronic acid in prostate cancer patients with bone metastasis (*n* = 30).	Once-weekly enteric-coated Orazol™ tablets containing 20 mg of zoledronic acid versus weekly Zometa^®^ (4 mg) i.v. infusion over 49 days.	Equivalent urine output biomarkers; claim of 5% bioavailability (BA) in patent.	[54]
Antisense oligonucleotide with C_10_ (ISIS 104838) in healthy subjects (*n* = 15).	Enteric-coated tablets, four formulations, and one after a high-fat meal. Subjects received all treatments.	9.5% bioavailability compared to s.c. No study-related adverse effects.	[58]
Basal insulin in C_10_ formulation versus insulin glargine in Type 2 diabetics (s.c.) (*n* = 25).	Daily tablets of a long-acting insulin (I338) over 8 weeks.	1.5–2.0% bioavailability compared to s.c. Comparable reductions in plasma glucose.	[55]
Insulin tregopil (IN-105) in C_10_ tablets in healthy subjects.	Single treatments of insulin along with metoformin over 4 periods of 2 days.	No effects on the pharmacokinetics (PK) of metformin; good safety.	[56]

LH, luteinizing hormone; FSH, follicle-stimulating hormone; s.c., sub-cutaneous; i.v., intravenous. The Phase II study [55] is the most comprehensive of these studies.

**Table 2 pharmaceutics-11-00078-t002:** Summary of data from selected studies in humans reported for a range of poorly permeable molecules formulated with either salcaprozate sodium (SNAC), monosodium *N*-(4-chlorosalicyloyl)-4-aminobutyrate (4-CNAB), or 8-(*N*-2-hydroxy-5-chloro-benzoyl)-amino-caprylic acid (5-CNAC). T2D—type 2 diabetes; sCT—salmon calcitonin.

Description	Treatment	Outcome	Reference
Vitamin B_12_ with SNAC in tablets in healthy subjects (*n* = 20). Medical food clinical study.	(A) Two tablets, each with 5 mg of vitamin B_12_ with 100 mg of SNAC (B) One tablet: 5 mg of vitamin B_12_ with 100 mg of SNAC (C) One commercial tablet: 5 mg of vitamin B_12_ (D) 1 mg of vitamin B_12_ via i.v. injection.	Treatment (B) achieved 3% higher absolute BA compared to the commercial oral formulation. No adverse effects.	[20]
Heparin with SNAC in hip replacement patients, (*n* = 123). Phase II.	Two studies: one dose every 8 h (max 16 doses), and two doses every 8 h (max 12 doses).	Achieved anti-factor Xa activity comparable to s.c. heparin. No change in major bleeding events compared to s.c.	[62]
Insulin with 4-CNAB in untreated T2D (*n* = 10). Phase II.	300 mg of insulin with 400 mg of 4-CNAB, or 15 IU of insulin s.c. Performed under fasting conditions.	C_max_ was higher and was reached faster compared to s.c. Shorter duration and high subject variability. No adverse effects.	[65]
sCT with 5-CNAC in osteoarthritic patients over 24 months (*n* = 1176 and *n* = 1030) Phase III.	0.8 mg of sCT in tablets twice daily for 24 months.	No significant effect compared to placebo.	[66]
sCT with 5-CNAC in postmenopausal women with osteoporosis (*n* = 4665). Phase III.	0.8 mg or placebo in tablets daily, together with vitamin D and calcium for 36 months.	No beneficial effect on fractures was observed. No change in quality of life.	[6]

**Table 3 pharmaceutics-11-00078-t003:** Selected clinical trial data with an emphasis on peer-reviewed literature from the daily semaglutide/SNAC oral tablet formulation from Novo Nordisk in T2D patients.

Description	Parameters	Comment	Reference
Phase II dose-ranging 26-week study in patients (*n* = 632) (NCT01923181).	0.7–1.9% reduction in glycated hemoglobin (HbA1c); some weight reduction; mild gastro-intestinal (GI) side effects common.	The key trial which supported moving to Phase III.	[69]
PIONEER-1 Phase IIIa 26-week study in patients (*n* = 703) (NCT02906930).	Mean 1.5% reduction in HbA1c confirmed with 14-mg dose; 4.1-kg weight reduction; mild–moderate nausea in 16% versus 6% in placebo.	14 mg established as semaglutide dose with 300 mg of SNAC in all studies.	[71]
PIONEER 5 Phase IIIa in renal-impaired patients (*n* = 71) (NCT02014259).	5 mg of semaglutide for 5 days; 10 mg for 5 days, assessed up to 21 days after; no change in PK overall.	Area under curve (AUC) and half-life (t_½_) similar to regular T2D patients, no need to change dose regime.	[72]
Trial in hepatic-impaired patients (*n* = 56) (NCT02016911).	Design as for PIONEER-5.	AUC, C_max_, and t_½_ unchanged, no need to change in dose regime.	[73]
Trial in healthy subjects ^1^ taking omeprazole (*n* = 54) (NCT02249871).	5 mg for 5 days, followed by 10 mg for 5 days) ± 40 mg omeprazole.	AUC and stomach pH slightly higher in semaglutide/omeprazole group, but no need to change dose regime.	[74]
PIONEER-6 Phase IIIa assessed cardiovascular (CV) risk in T2D patients (*n* = 3183) (NCT02692716).	Primary end-points: reduction in major CV events over median 16-month period.	Cardiovascular (CV) outcomes not different from placebo, but suggestion of a mortality benefit of oral tablet.	[75]

^1.^ All studies in T2D patients except for the omeprazole study. ^2.^ PIONEER Phase III 10 study designs are available at https://pharmaintelligence.informa.com/resources/product-content/novos-oral-semaglutide-passes-pioneer-2-but-weight-loss-result-a-bit-disappointing (accessed 12 February 2019). Details of all oral semaglutide trials are available at www.clinicaltrials.gov.

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
