# Peer review of "Intestinal Permeation Enhancers for Oral Delivery of Macromolecules: A Comparison between Salcaprozate Sodium (SNAC) and Sodium Caprate (C10)"

_pharmaceutics, 2019, doi:10.3390/pharmaceutics11020078_

Round 1
Reviewer 1 Report
JOURNAL: Pharmaceutics
MANUSCRIPT #: 440891
SUMMARY:
In the manuscript, the authors provide a balanced historical and comprehensive review of salcaprozate sodium (SNAC) and sodium caprate (C10), two of the most studied and efficacious chemical permeation enhancers characterized to date. Specifically, the review addresses their mechanism of action, clinical trial evaluation and regulatory perspectives. This is a timely piece which will likely be of broad interest particularly given the increasing attention and efforts towards delivery of macromolecules orally, underscored by the recent successes of the PIONEER trials.
The review is well organized and written covering: challenges for oral delivery of macromolecules, proposed mechanisms of C10 and SNAC based on our current understanding (including para/trans-cellular permeability, gastric fluid neutralization, performance comparison, safety evaluation, and efficacy of delivery).
I have no doubt that this timely piece will be of interest to scientists in academia and industry. A few comments that should be addressed are listed below.
Minor:
-introduction: for completeness please include other routes of administration in “Over 90% of peptides are administered as injectable, with just 4% delivered orally i.e. intranasal, Inhalation, Transdermal, etc)
-although the review focuses onC10and SNAC, brief mention and referencing of other permeation enhancers should be noted for completeness (e.g, Advanced Drug Delivery Reviews 106 (2016) 277–319).
-P2. L38-change “are administered as injectable” to “are injected”
-would include the following reference and also review briefly for completeness given use of GIPET - Halberg, I.B., Lyby, K., Wassermann, K., Heise, T., Zijlstra, E., and Plum-Mörschel, L. (2019). Efficacy and safety of oral basal insulin versus subcutaneous insulin glargine in type 2 diabetes: a randomised, double-blind, phase 2 trial. Lancet Diabetes Endocrinol.
-P.24 – change: trails to trials
-conflicts of interests: complete names of entities providing consulting fees, grants, royalties, equity should be provided
Author Response
Referee 1
Minor:
1. Introduction: for completeness please include other routes of administration in “Over 90% of peptides are administered as injectable, with just 4% delivered orally i.e. intranasal, Inhalation, Transdermal, etc)
Response: We have expanded on material from reference [2] to encompass data on other routes.
2. Although the review focuses on C10 and SNAC, brief mention and referencing of other permeation enhancers should be noted for completeness (e.g, Advanced Drug Delivery Reviews 106 (2016) 277–319).
Response: This reference has been cited within and is expanded upon.
3 P2. L38-change “are administered as injectable” to “are injected”
Response: this has been changed
4. Would include the following reference and also review briefly for completeness given use of GIPET - Halberg, I.B., Lyby, K., Wassermann, K., Heise, T., Zijlstra, E., and Plum-Mörschel, L. (2019). Efficacy and safety of oral basal insulin versus subcutaneous insulin glargine in type 2 diabetes: a randomised, double-blind, phase 2 trial. Lancet Diabetes Endocrinol.
Response: Yes, that trial was published while the paper was under review and we have now cited it in the C10 table, along with a new reference on C10 in clinical trials with Biocon’s insulin (also has C10 in the formulation). We thank the Reviewer for pointing this out.
5. P.24 – change: trails to trials
Response: Completed
6. Conflicts of interests: complete names of entities providing consulting fees, grants, royalties, equity should be provided.
Response: This has been provided now in more detail.

Reviewer 2 Report
Salcaprozate sodium (SNAC) and sodium caprate (C10) are two of the most advanced intestinal permeation enhancers (PEs) that have been examined in clinical trials for oral delivery of macromolecules. Their effects on intestinal epithelia have been studied for over 30 years, yet there is still debate over their mechanisms of action. Therefore, the authors tried to investigate the ability as well as toxicity as PEs for macromolecules of two surfactants based on a lot of papers published in the last 20-30 years.
Overall, the manuscript is written very well, and comparison of the SNAC and C10 in various aspects has been described in excruciating detail. It is considered that this review would highly useful for the readers who are interested in “mucosal absorption enhancers” and “development non-invasive delivery systems for macromolecules”. However, the reviewer is wondering how many general readers would like to read this too long review specialized only for the substance SNAC and C10. I am wondering whether general readers can get width range of knowledge about PEs from this review. It may be worth as the historical record of the 2 unique surfactants, however, the reviewer can not find valuable scientific messages throughout the manuscript. Therefore, the reviewer recommends to condensate more the content throughout the manuscript, and to show clear fruitful insight of PEs for readers.
Regarding 6 (How do SNAC and C10 alter GI permeability?)
In this section, it is good that the authors pointed out the various points of view and the problems in evaluation to solve the mechanism of action of the absorption promoters. Clarification of mechanism of action of PEs is quite important to avoid unexpected toxicities in clinical. In addition, it is valuable for critically examining both surfactants in detail, and also good to pose problems, however, anyway it is too redundant. It is not clear what the authors ultimately want to tell the readers. The contents should be summarized more and show the message for the readers clearly.
Regarding 7 (C10 and SNAC: pharmacokinetics and efficacy in clinical trials)
In this section, for the Clinical study conducted on these substances, selected tests are listed in Tables 1 and 2, and the rest are described in sentences. Those clinical studies are written in a very detailed, and this section is seemed just only a record of the results of the completed clinical trials for specific readers. It is necessary to considerably condense using Tables. Then, if it is needed to point out, the authors should discuss the difference in their clinical effects, usefulness, and/or problems with relevance to the clinical studies.
The reviewer thinks that the final sentence says too much. The authors should give discussion fair, patients would take a product from various options on their preference.
Author Response
Referee 2
1. However, the reviewer is wondering how many general readers would like to read this too long review specialized only for the substance SNAC and C10. I am wondering whether general readers can get width range of knowledge about PEs from this review. It may be worth as the historical record of the 2 unique surfactants, however, the reviewer cannot find valuable scientific messages throughout the manuscript. Therefore, the reviewer recommends to condensate more the content throughout the manuscript, and to show clear fruitful insight of PEs for readers.
Response: We have now condensed in some areas (marked), noting that it is not a general PE Review. We already did that more general exercise for ADDR (Maher, S et al. 2016). The historical record for C10 and SNAC needed to be revaluated in terms of the new clinical data around these two agents. It is a very precise aim, so we have brought it out better so that there is no confusion.
2. Regarding 6 (How do SNAC and C10 alter GI permeability?). In this section, it is good that the authors pointed out the various points of view and the problems in evaluation to solve the mechanism of action of the absorption promoters. Clarification of mechanism of action of PEs is quite important to avoid unexpected toxicities in clinical. In addition, it is valuable for critically examining both surfactants in detail, and also good to pose problems, however, anyway it is too redundant. It is not clear what the authors ultimately want to tell the readers. The contents should be summarized more and show the message for the readers clearly.
Response: We have put in a conclusion line on the in vitro data, but the message is that the data is contradictory in places; in addition mechanisms shown in vitro do not necessarily hold up in vivo due to the high doses and altered GI milieu in different part of the gut.
3. Regarding 7 (C10 and SNAC: pharmacokinetics and efficacy in clinical trials) . In this section, for the Clinical study conducted on these substances, selected tests are listed in Tables 1 and 2, and the rest are described in sentences. Those clinical studies are written in a very detailed, and this section is seemed just only a record of the results of the completed clinical trials for specific readers. It is necessary to considerably condense using Tables. Then, if it is needed to point out, the authors should discuss the difference in their clinical effects, usefulness, and/or problems with relevance to the clinical studies.
Response: We have edited the text a little in respect of some detail and refer them more to the tables in order to condense. Many readers will not be aware of this older literature, hence the detail is important and informs current trials. However, many of these studies are hard to compare objectively due to differences in designs, formulations, and read-outs.
4. The reviewer thinks that the final sentence says too much. The authors should give discussion fair, patients would take a product from various options on their preference.
Response: It is a bit difficult out work out what the Reviewer is trying to say here. In the last sentence, we merely point out that there are restrictions and inconveniences around eating and drinking that may influence patient preferences and this may impact the ultimate success of daily oral semaglutide product; such inconvenience is also seen for once-weekly oral bisphosphates. We have amended that sentence accordingly.